# A Comedian in the Pulpit: Empowering the Use of Humor in Preaching

Philmont Devon Bostic

School of Ministry, Palm Beach Atlantic University, West Palm Beach, FL 33401, USA; bostipd@pba.edu

**Abstract:** Each week, the preacher mounts the pulpit with many tools to deliver an impactful sermon. One element of preaching that the black church should embrace is humor. Humor and preaching may appear strange bedfellows, but humor is embedded in the art of black preaching. This study explores humor within the confines of the African Methodist Episcopal Zion Church.

**Keywords:** black church; humor; Methodist





## 1. Introduction

Preachers spend hours crafting a sermon each week. They search for the right words and meanings and tie them together to help their congregation understand God's Word. There are books to help teach pastors how to exegete the text correctly. There are commentaries to ensure preachers understand the context and meaning behind what they are reading. There are books to teach the proper use of their voice to drive home a point. There are even books that illustrate how to celebrate in preaching. However, very little is offered to preachers to invoke humor into their sermons and the worship experience. There has been resistance throughout the years to fully embrace the use of humor as something that is not only appropriate but necessary for a complete experience with God. Bernard Schweizer suggests, "Laughter and faith have a vexed relationship with one another: laughter tends to pull us in the direction of irreverence and subversion while (monotheistic) faith tends to pull us in the direction of irreverence" (Schweizer 2017, p. 135). This study explores humor and comedy in preaching within the black church context. This will be completed by employing the four-fold word pair method.

## 2. Theological Research Problem

This study is needed for several reasons. There needs to be more work being performed on the benefits of humor in a group setting. Some studies look at the health benefits of humor. Much work has been performed on the psychological benefits of humor. Humor can help lift a person's emotions when they are down. Humor can also help raise one's psychological health. Cann, Stillwell, and Taku state, "Thus, it would appear that good humor can support maintaining a stable positive personality style, which has positive associations with both psychological and physical wellbeing" (Cann et al. 2010, p. 213). However, these articles reveal the benefits in a group setting; there is a limited review of these benefits within a church setting.

Secondly, much work showcases the connection between the black church and West Africa. The exploration of humor and the black church is virtually nonexistent. The study will explore the shared link between the black church and black humor. Both have initial connections that will be explored in greater depth later. The black Church and black humor draw their origins to West Africa. This study will examine how West Africans carried with them through the middle passage their religion and style of humor. The study will explore the connection between the oral tradition of black spirituality and humor.

Thirdly, the study will explore the connection between humor and The Bible. The study will also explore the relationship between humor and the biblical text. Through

exegesis of the crucial text, the study will look to link to humor and worship in The Bible. This will be achieved by comparing the three texts in the Bible where God is said to laugh in the Psalms, as well as Old and New Testament texts that reflect humor in worship.

The study will focus on three key questions. First, does humor have any place in preaching in the black church? The second is how humor is essential in an African-American church context. Thirdly, is there genuine theological support for black comedy?

With the focus on the black church, the study will further narrow its focus to the use of humor in the African Methodist Episcopal Zion Church (AMEZ). This research is partially ethnographic, as the author served as a pastor in this context for over a decade. This church is a part of the Black church experience and begins in America. This church will help focus on a specific culture—black culture. Black culture is a rich culture that needs further exploration regarding the use of humor and worship.

### 3. Thesis

To answer the previously stated questions, this study suggests that the AMEZ looks to employ a few concepts. The first is a greater understanding of the church's history within the black church tradition, also by the preacher engaging in black comical theology. The study suggests that the use of these two concepts will bring about the use of humor in the black church, which can be fulfilled if the black preacher understands his role in the black tradition. Then, by employing a black comical theology, these ideas fall in line with traditional black theology. Frederick Ware points out three different schools of Black theology in his book. The Black comical theology would combine The Black Hermeneutical School and the Human Science Schools. The Black Hermeneutical School focuses on liberation, while the Human Sciences School focuses on empowerment (Ware 2012, p. XVI). A theology of black comedy looks to empower the person while seeking to use humor to liberate them first from their circumstances.

Black comical theology is a theology that looks at the work of God in the lives of the oppressed. The black comical theologian uses humor to understand God and themselves in a better way. Black comical theology understands the power of oppression and turns it on its head. This view of God recognizes that God is a God of reversal. Followers of this theology comprehend that in the end, no matter what happens, God will reverse all the wrongs in the world. This theology applies to our lives now as well as the future.

Black theology began taking shape during the civil rights movement in the 1960s in America, where black people wrestled with the ideas of God and how to handle the oppressive nature in which African Americans found themselves. Ware states, "The first stage of the emergence of black theology started with the civil rights and black power movements" (Ware 2012, p. 2). With the Black hermeneutical school, theologians promoted the idea of liberation. Black comical theology builds on this idea and takes it forward by seeking liberation and looking to empower, such as in the Human Services School. People need physical and spiritual liberation but must also be assigned to go ahead after finding that freedom. Using humor both liberates and empowers.

A historical analysis of the plight of African Americans is full of oppression. The implementation of black comical theology seeks liberation and empowerment. They used it to liberate their minds from the horrors of slavery and oppression. As will be discussed later, black people laughed through hardships. They also used humor to empower themselves from the ideas brought on by slavery. They used humor to point out the faults in social and church views on the status of African Americans. This theology can still work today, as people use God and his work in the community to liberate people from their spiritual bonds. Black people use humor to showcase fault and help to uplift. Because of these ideas, humor is the best way to help black preachers engage their congregation.

### 4. Literary Review

For African Americans, humor is embedded within the culture. African Americans have used humor to combat the social and political ills of slavery and oppression. One of

the first ways this was accomplished was by living a double life. The oppressed would have one way in front of their oppressors and another behind closed doors. This is seen in William Schechter's book; he states, "Thus did a black man scarcely removed from African origins summarize the function of humor in the Afro-American history: a balm against oppression".

Upon arrival from West Africa, African Americans brought a culture of humor. West Africans had stories and tales of tricksters. In many West African creation narratives, there were stories of twins who used trickery to get what they wanted. Will Coleman mentions the twins Zinsu and Sagbo, who were trickster spirits. Coleman says, "There are several stories of how Zinsu barely escapes being eaten by the dreaded yehwe (a type of predatory vodun), which are thirty horned monsters who live in the forest" (Coleman 2000, p. 25). The priest or priestess would tell these stories. This person also served as the griot. Coleman says, "He or she was also often the griot (storyteller and chronicler of familial and tribal traditions), fortune-teller, and healer within the community" (Ibid, p. 33).

Coming to America, these stories were modified. The trickster narratives were told to help fight oppression. The trickster used humor to combat the ills of slavery and oppression. This spawns the use of folk tales, such as Br'er Rabbit and other stories. Schechter states, "They were able to rid themselves of their aggressions by overly gloating over the torture and death of the fox. The harmless scary rabbit was converted into a superhuman myth hero who could do and say all the things black could not" (Schechter 1970, p. 29). They would point out the fault of the oppressor without them knowing it. This was a skill that is known as signification.

Signification is a skill that began in Africa but was further developed during slavery and has carried over into modern times. The *St. James Encyclopedia of Hip Hop Culture* defines signification as "a rhetorical technique in the black vernacular characterized by verbal indirection and innuendo" (Rooks 2018). This is popular in West African tales of the signifying monkey. The use of a monkey is similar to that of the Br'er Rabbit, and the monkey has been used as a term of degradation for African Americans. Henry Louis Gates says, "The ironic reversal of a received racist image of the black as simianlike, the Signifying Monkey—he who dwells the margins of discourse, ever punning, ever troping, ever embodying the ambiguities of language—is our trope for repeating and simultaneously reversing in one deft, discursive act" (Gates 1983, p. 686). The basis for African American Humor is pointing at the incongruities of life and awaiting the glorious reversal.

Reversal is an essential aspect of African-American humor. It follows the Theory of Humor. This Theory states that jokes are implied when the ending is not congruent with the beginning. Incongruity Theory occurs when situations do not turn out how we would believe (Kulka 2007). Philosophers such as James Beattie, Immanuel Kant, and Kierkegaard were proponents of the idea. This is how African Americans base their theology that God is working towards the reversal of the oppressed.

These uses of humor are tied to black theology. Many African American preachers lean into the reversal work of God—especially the reversal work of God on the cross. Christ is on the cross, and his oppressors are laughing at him. M.A. Screech points out, "Laugher is one of the ways in which crowds, thoughtless, cruel or wicked, may react to the sight of suffering" (Screech 1999, p. 17). African Americans experienced similar laughter as crowds showed up to witness lynchings in America. James Cone saw the similarity of Jesus on the cross to people being lynched. Cones states, "That God could 'make a way out of no way' in Jesus' cross was truly absurd to the intellect, yet profoundly real in the souls of black folks" (Cone 2011, p. 2). Black people understand a messiah who has suffered like them and understands their pain. When the African American preacher stands in the pulpit, they are standing on the traditions of the trickster in the African American culture.

## 5. Methodology

The four-fold word pair method will work as follows. The study will first begin by inserting and identifying the subject. The subject is AMEZ. The study will determine the

church outline in NancyAmmerman's book *Pillars of Faith: American congregations and their partners*. The study will focus on the black church overall. Next, the study will focus on assessing and analyzing and will begin by reviewing the history of the AMEZ. It will then present the statistics collected from the survey given to pastors from the AMEZ. The study will move to correlate and confront and will examine how the church can look to apply humor in the church and then deal with the challenges of applying humor. Last, the study will explore expanding and empowering. This section will expand further on the uses of humor whilst also examining the connections or disconnects of humor.

### 5.1. Inserting–Identifying

The focus of this study will be on the African Methodist Episcopal Zion Church. The AMEZ traces its history to the John Street church in New York in 1796 (Miller 1963, p. 2). The John Street church was one of the oldest Methodist churches in America. The church initially welcomed black members, but the black congregants were pushed to the side after new members joined (Hood 1987, p. 1). The black congregants were allowed to hold their meetings shortly after. In his book, *One Hundred Years of the African Methodist Episcopal Zion Church*, Bishop J.W. Hood states, "These meetings were regarded as prayer meetings, but the leaders frequently gave exhortations—in fact, did such preaching as their abilities permitted" (Ibid, p. 57). This continued for another 20 years until a breaking point was reached. The ordination of preachers caused a complete separation between the black congregation and the denomination. Other churches could eventually ordain their clergy, but in Methodism, the Bishop performed ordination at the General Conference. Finally, the members saw they would not gain the opportunity to fulfill their calling by God to preach, so they severed their agreement with the Methodist church after witnessing Richard Allen from the African Methodist Episcopal Church and Peter Spencer from the Union Church of Africans. Hood suggests, "If they had agreed to ordain a few of our men before 1813, there would have been one African Methodist Episcopal Church, of which old Zion would have been the fountain head" (Ibid, p. 62). Now, according to the General Secretary Auditor's office, this church has 3700 congregations around the globe.[1]

This church is very much a part of what is known as the Black Church. The Black Church was born in America. The Black Church separated itself from other churches in America around the early 1800s. Hood states, "Secessions from churches are generally the result of differences of opinions on doctrine or church government" (Hood 1987, p. 1). The Black Church was a significant movement in ending slavery and the civil rights movement. The Black Church has its feel and action that differs from other churches. Most of this comes from the church's ties to West Africa.

Furthermore, both within and alongside these more African belief systems, this African Diaspora developed unique forms of African American Christianity, just as ("pagan") Europeans and Euro-Americans had progressively (often with some ecclesiastical coercion) indigenized Middle Eastern and North African forms of Christianity, Greek philosophy, and Jewish thought centuries earlier through the formation of various Orthodox traditions, Roman Catholic orders, Protestant denominations, esoteric societies, and fraternal orders (Coleman 2000, p. 36).

### 5.2. Assessing-Analyzing

The Methodists appear to be serious people. You can see this by looking at John Wesley, the founder of Methodism, who rules how people should conduct themselves. In his four rules, Wesley states in rule two, "2. To labor after continual seriousness, not willingly indulging myself in the least levity of behavior or laughter; no, not for a moment" (Wesley n.d., p. 49). Though the AMEZ is a part of the Methodist movement, it still draws heavily from the black church tradition.

A survey was conducted to understand the beliefs of the preachers in the AMEZ. The author contacted 10 members of the AMEZ church for the survey. The survey was conducted via Google Forms. The survey was limited due to the availability of the preachers

within the AMEZ. Each person was sent a link to complete the survey. Google Forms tallied all the information gathered (Gordon 1998). This research gave a fundamental idea of the church's view on humor. The reason for a more comprehensive study was due to time and availability. These people were chosen due to personal connections and the belief in an unbiased answer. The participants were told to be honest and answer from their perspectives.

The survey contains six questions—each question after the first was graded on a scale of one through four. Level one was do not agree, going up to level four with strongly agree to be the highest. Each person was emailed the survey and instructed to answer honestly. Each person understood that the survey would be used for this study. The first question assessed the length of time the individual spent preaching. The following questions were to gauge how each preacher saw the use of humor in preaching and its relationship to the black church.

The purpose was to gauge each preacher's view on the participation of humor in worship, especially in preaching. The survey produced the following results. Of the preachers surveyed, 80% had at least 10 years of preaching experience. When asked whether they felt humor had a place in preaching, all the respondents strongly agreed. The next question asked whether those surveyed purposely used humor in preaching; 60% slightly agreed, leaving the other 40% somewhat agreeing. When asked if they feel the need to open or close their sermon with a joke, 40% said they do not agree, while 60% said they somewhat disagree. When asked whether leadership supported humor in preaching, 40% said somewhat agreed, while 60% said strongly agreed. Finally, when asked whether they felt that humor was part of the black preaching experience, 20% somewhat disagreed, while 40% agreed and strongly agreed.

### 5.3. Correlating and Confronting

The African-American experience in the country drives the humor of the people. Dexter B. Gordon states, "American slavery provides the backdrop of tragedy against which African Americans developed their distinct form of humor, in which material of tragedy was converted into comedy, including the absurd" (Dance 1988, p. 125). This form of laughter appears to be paradoxical. Black people laugh at things that should bring them to tears. Daryl C. Dance speaks to this by pointing out that during slavery, the happy narrative comes from this reality. She quotes John Little as saying, "They say slaves are happy because they laugh and are merry. I and three or four others have received two hundred lashes in a day and our feet in fetters, yet at night, we would sing and dance and make others laugh at the rattling of our chains" (Pang 2009).

The humor of African Americans is dark humor. It is similar in many ways to the humor of Jews. The Jews and Blacks carry a similar history of trauma and, in turn, similar uses of humor. Sander Gilman says, "Both black and Jewish humor is rooted in oppression culture, where people laugh at themselves to deal with adversity" (Hood 1987, p. 15). Both communities find humor in things that would appear to break others.

### 5.4. Expanding and Empowering

African Americans understood that humor was not only for survival but also a tool to help change their situation. Much of black humor is used to point out the hypocrisies of their system. They used humor as a tool of subversion. As stated earlier, during slavery, they would use stories such as Br'er Rabbit to point out the hypocritical views of those in power.

This history is found in the AMEZ. Two of the more famous church members are Fredrick Douglass and Sojourner Truth. Both were members of AMEZ. Hood says of Douglass, "Fred Douglass, one of the most remarkable men that the race has produced, admits that he is indebted to the African Methodist Episcopal Zion Church in Bradford, Mass. for what he is" (Ganter 2003, p. 535). Though the abolitionist is known for his fiery speeches, such as his speech on the Fourth of July, he was also a humorist.

Fredrick Douglass used humor as a technique to help in debates and other activities, but Douglass was careful in how he used humor. A popular form of entertainment at the time was that of minstrel shows. Many of these minstrel shows were used to debase black people and perpetuate stereotypes. Douglass walked the line between that but used some elements to support his point. Granville Ganter says of Douglass, "By exploiting his audiences' likely prejudices, however, Douglass used humor to transform himself from a social pariah into an equal" (Goldner 2012, p. 50).

Sojourner Truth also used humor to her advantage in a debate with a young lawyer about the place of African Americans in America, especially black women. Truth's debater alluded to many African American women being domestics. Sojourner Truth waited for her opponent to finish and retorted that she did not mind doing the dirty work of social change. She pointed out that she was likely the perfect person for it. Pointing to her time as a slave, she understood that black women have always been willing to do the dirty work.[2]

More recent examples of humor in preaching can be found in the two members of the AMEZ. In the AMEZ Church, there were a few examples of preaching that engaged in humor. Preachers often quoted famous sayings by comedians during the sermon. One example of this comes from Bishop W. Darin Moore. Bishop Moore is a sitting Bishop in the AMEZ. Bishop Moore would often quote the famous comedian Jackie Moms Mabley. Moore would say, "To quote that great theologian, Moms Mabley, if you always do what you've always done, you always get what you always got". Moore would use this quote in preaching to spur change in the congregation.[3]

Preachers would often close or open their sermons with a humorous story. This would be done to drive home the point. During a conversation with Rev. Dr. David T. Miller, he explained how humor can be used in a sermon to challenge the status quo. Dr. Miller was the pastor of a church in Vallejo, California, near Silicon Valley. Dr. Miller had once told a story to his Kyle Temple AMEZ Church in 2014. It is a story of people flying in a small plane. The pilot, a computer nerd, a cub scout, and a preacher were on the flight. While on the plane, they encountered severe turbulence, so much so that the pilot knew the plane was crashing. The pilot looked at the passengers and explained that he did everything possible, but the plane would crash. "There are only three parachutes and being that I have a wife and three kids, and they need me, you have to decide who gets the other two". The pastor hears this and begins to pray, and when he opens his eyes, the computer geek says, "Sorry, pastor, but while you were praying, I grabbed the other parachute. The world needs me as a computer genius," so he took the parachute and jumped. The pastor looks at the cub scout and says, "Son, I have lived a good life; you can take the parachute. I have prayed and made my peace with God". The cub scout says, "Oh, no, sir, there are two parachutes left". He said, "How do you figure, son?" "The pilot took one, and the computer guy took the other, the cub scout said, "he didn't take a parachute; he took my book bag" (Coleman 2000, p. 102).

## 6. Findings

### 6.1. The Black Church and Comedy

The AMEZ has ties to comedy, but the tradition runs throughout the black church. Both come from West African traditions that expanded and grew under the weight of slavery. Black people took the narratives given to them in both cases and used them for their liberation. If we first begin by looking at the Black Church, we will see that there was not much activity until after slavery ended. Will Coleman states, "According to C.B. Burton, African Americans had no church of their own until the end of the Civil War" (Walker 2015, p. 11). This does not mean all black churches were formed officially until after the Civil War. There were black churches, such as the AMEZ, that were moving toward formation before the war, but after the war, the church began to expand. The common thread between the black churches is dealing with oppression.

Much of the slave narratives that Coleman and others reviewed demonstrate how African Americans gravitated towards Christianity. Though it was illegal for the enslaved

people to read, they would often be told the stories of The Bible. Many of the enslaved people gravitated toward the story of the Hebrew exodus. This story and others were removed from the "slave bible", a redacted version of scriptures used to help with the oppression of the enslaved people. The slaves saw through this and saw themselves in the story of the Jews. The same God who brought them out of bondage is the same God who could do the same for them. Abolitionist *David Walker's Appeal* spoke about Moses wanting to side with the oppressed over the oppressor. In his allusion to Exodus 2, Walker says of Moses, "But he had rather suffer shame, with the people of God that enjoy pleasures with that wicked people for a season" (Abrahams 1962, p. 209). Ideas like Walker's helped the enslaved see their value and that they were on the right side of God.

It was during the time of slavery that black comedy began to take root. During this time, African Americans had little way to fight back against their oppressors physically. So, they fought back mentally. They first began by touching themselves mentally by playing the dozens. This is a mental and verbal game where African Americans take turns battling each other through insults. Roger Abrahams defines the activity this way, "One insults a member of another's family; others in the group make disapproving sounds to spur on the coming exchange" (Coleman 2000, p. 91). This verbal jousting is the training young people engage in that not only provides them with a way of relief through humor but sharpens the verbal and mental sharpness to become a trickster.

As stated earlier, the trickster in African-American culture is a person who uses his mental skills and capabilities to help point out the hypocrisies of a situation. One main way the tricker engaged in this activity is through signifying. The trickster would use many forms of language to get their point across and to point out flaws in the logic of the oppressor. Trickster tales can be found in African-American folklore.

Another trickster points to the signifying monkey. He used a creature that was a common racial epithet towards black people as the hero. The monkey would often trick the lion and get out of trouble because of its wit and skill. However, the lion was strong physically. He could not match the wit of the monkey. These stories and others, such as Br'er rabbit, were often used to explain the incongruities of slavery and the hypocrisy of the master but told in a way that the master never caught on that they were the butt of the jokes. These skills were also used in black preaching.

Black preaching often took place on the plantations of the South. The slave owner often tasked the black preacher to preach to the enslaved. In this lies a paradox of slavery. A black preacher was to preach to enslaved people that their place was enslavement and to maintain the proper decorum to gain liberty in the next life. The freedom that Jesus brought was only for those of a certain race. Though they were perceived to care for the soul of the enslaved, they cared little for the body of the enslaved. Coleman points out, "Thus, Southern Christians cult(ure) exists within the heretical paradox of economic decency, white supremacy, and evangelical sensibilities" (Mills 2015, p. 44). The slave owners sought to use the black preacher to help maintain civility and to have the slave accept their place in life.

The black preacher saw through this ruse and used the time allotted for preaching to showcase the hypocrisy in the slaver's thought process. Zachary Mills states, "Some black preachers often used the sermon as a highly coded message to be grasped by 'insiders' and though intelligible to the English speaker, often remained out of the full reach of 'outsiders'" (Ibid, p. 2). They relied on their trickster abilities to live this double life.

*6.2. How Is Humor Overlooked*

At first glance, humor would be an important element in preaching in the black church, but it is so embedded that it is often taken for granted. When looking at black preaching, when preaching is doing a great job, another person might say that the preacher is acting a fool. In other contexts, this would appear to be an insult, but this is a compliment in black preaching. Acting or behaving "foolish" in this context is fully embracing what you are doing. Zachary Mills states, "During especially emotional, climactic moments during a

sermon, congregants or fellow preachers will often say of the person preaching, 'She's a fool!' Or, overwhelmed with the truth and relevance of a sermon, others will exclaim, 'That fool is preaching!' These statements are not meant to disparage a preacher" (Levine 2007, p. 327). Because of the "foolish" nature of black preaching, the preachers might overlook this as a skill. The fear of many black preachers is becoming a trope or expression.

Many black preachers look to fight the common tropes of the black preacher. Most comedians from Richard Pryor, Eddie Murphy, and beyond have comedic renditions of the black preacher. The black preacher is viewed as loud, boisterous, and often phony. Lawrence Levine points out: "The substantial anger black felt at the hypocrisy on top of the figure of the black minister, whose lofty pretensions were constantly pictured as being undermined by his compulsive lust for chicken, liquor, money, and women" (Mills 2015, p. 38). The black preacher would preach holiness but live a double life. Many comedians pick up on this trope and have presented it for years. In the movie *Car Wash*, Richard Pryor plays a man who could be a pimp or a preacher. The point is that the lives of both closely resembled each other. If preachers knew these tropes, the goal would be to push back against this narrative. These tropes were used to invoke humor, and the black preacher would look to avoid being labeled in this way.

## 7. Theological View of Humor

As mentioned earlier, there are few examples of humor in The Bible, but there are ways to look at The Bible considering humor. Mills offers two examples of this: one is Inversion/Reversal, and the other is indirection. These skills come from the African-American trickster tradition. These ideas can be used further to understand the theological views of African Americans and humor. First, the thought of inversion or reversal. As Mills states, "This practice involves turning normative expectations and categories on their heads, inverting them, in order to establish a context in which a new meaning can be experienced" (Cone 2011, p. 2). This can be seen all through The Bible. An Old Testament would be from Exodus 14:19–31. God called on Moses to leave Egypt, and the people were trapped with the sea before them, with the Pharoah's army chasing them down. All-natural measures stuck the people, and their exodus from Egypt was now being thwarted. Still, God reversed the situation by having Moses part the Red Sea, and the children of Israel crossed onto dry land, drowning the Pharoah's mighty army. This bit of scripture is popular within the black church because it reminds the people that their situation can change in a heartbeat with God on their side.

As mentioned earlier in the New Testament, the ultimate example of reversal is that of Jesus on the cross. Much like the Exodus story, Jesus' situation was over. He was taken to the cross and killed by the Roman government. They assumed that placing him on the cross would end his movement, but God used the cross to fix the brokenness in the world. James Cone saw black people's connection to the cross's reversal. Cone states, "That God could 'make a way out of no way' in Jesus' cross was truly absurd to the intellect, yet profoundly real in the souls of black folks" (Mills 2015, p. 41).

Now, we shift our focus to the other skill of the black preacher: indirection. Mills says of indirection, "Indirection involves the communication of message through an intentionally subtle, cryptic or roundabout way" (Goenawan 2021, p. 50). In many ways, this is one of the more used skills of the black preacher—a way of talking about something without talking about it. When it came to lynching in the American South, a popular term was "strange fruit". This term comes from a poem by a Jewish man named Abel Meeropol. (Ali 2008). The poem was popularized by the singer Billie Holiday. The song speaks about lynching victims as strange fruit. Jesus could be viewed as a strange fruit hanging from the cross. When black preachers reference Jesus being lynched, it has a deeper meaning to the black soul. Cone says, "Like the lynching tree in America, the cross in the time of Jesus was the most 'barbaric form of execution of the utmost cruelty' the absolute opposite of the human value system" (Cone 2011, p. 35).

## 8. Discussion

### 8.1. Humor Reimagined

New skills must be acquired for the preacher to engage in humor fully. The first step would be to have a different understanding of laughter. As stated earlier, only three scriptures demonstrate God laughing: Psalms 59:8, Psalms 2:4, and Psalms 37. Jacqueline Bussie points out that these texts show God laughing at man's hubris. This means that when the oppressed people laugh, they are laughing with God. God's laughter in this text reflects God's knowledge of how things will turn out. When oppressed people laugh, they laugh, understanding that God will work this all out. Bussie states, "A theology of hope must be the counterpart of a theology of laughter" (Bussie 2007, p. 184). When those facing oppression laugh, they laugh with the hope of a brighter tomorrow.

This falls in line with the tradition of the black preacher. The black preacher must fully embrace their role as a trickster. There are elements in black preaching that go back to the trickster role. The trickster used humor to showcase hypocrisy and to initiate liberation. The preacher in the AMEZ should seek to do the same. This does not mean the preacher operates as sociologist and comedy historian Mel Watkin suggests as a clown (Hansen 1994, p. 1). The role of the clown has some success, such as minstrel actor Bert Williams, but the more effective form of humor is satire. The preacher should find ways through humor to challenge and empower their congregations.

This could be achieved by unifying the work of a trickster with a new view of scripture. This combination would be a black comical theology that combines the work of Jacqueline Bussie and black theology. Bussie's work helps fill the gaps where other theologians of laughter tend to fall short. Their work seeks to demonstrate the permission of laughter but does not cover the areas of laughter where laughter seems to fall short. Schweizer says, "Thus theologians of laughter such as Kuschel and Arbuckle condemn the superiority laughter which manifests itself in mockery and scorn, they fail to recognize that 'inferiority laughter', i.e., laughter issuing from a place of disempowerment and oppression, also tend to take on bitter, mocking, and sardonic tone" (Schweizer 2017, p. 139). This is how laughter must be separated between good laughter and bad laughter. Evil laughter mocks and demeans, such as with those who laughed at the foot of the cross. Good and Holy laughter is the laughter of oppression, knowing God will fix it.

### 8.2. Conclusions

This study demonstrates the potential for humor within the AMEZ. The research suggests that the black church and humor have had a great relationship within their shared history. From the humor of slaves on the ship through chattel slavery and Jim Crow humor, the church has worked together. The work performed here is the beginning of a larger conversation about the work of humor and homiletics within the African-American context. The implementation of the black comical theology would allow better engagement with homiletics. The poll conducted demonstrates the potential for greater concentration in humor in preaching. The work above also shows potential for greater attention in the historical history of humor within the AMEZ. There is still a greater need for exploration in the work of the black church and humor, but there appears to be some resistance to humor.

Other areas within the context of humor could be explored, reflecting greater exploration in other areas. First, a further study in other denominations use of humor. During the research process, it was discovered that the Eastern Orthodox Church has a service called "Bright Sunday". This worship service is conducted the Sunday after Easter. The service is to commemorate God's reversal of Satan's plan in the death of Jesus. A further study in this service could reveal more ways humor could be employed within the AMEZ.

Secondly, the poll conducted amongst the preachers was beneficial but led to a need for further expansion. While the answers benefited the work here, they showcased the potential for additional work across the AMEZ and the other Black Methodist denominations. As stated earlier, this work opens the door to more significant research.

Finally, humor within the context of theology is a severely underexplored area. There are volumes of work from a historical, sociological, and even philosophical view, but humor and theology have limited voices. These voices are almost nonexistent within the realm of the black church. As showcased above, humor in the black church was born out of slavery and has helped sustain African Americans throughout the centuries, but little work has demonstrated the potential and power of humor in theological formation as well as spiritual growth. This work could be the first step in the expansion of the field.

**Funding:** This research received no external funding.

**Institutional Review Board Statement:** Not applicable.

**Informed Consent Statement:** Not applicable.

**Data Availability Statement:** Not applicable.

**Conflicts of Interest:** The author declares no conflict of interest.

## Notes

1　　https://www.ameziongsa.com/, accessed on 22 June 2023.
2　　The author was in the congregation for this sermon circa 2019 in Johnson City, Tennessee.
3　　The author spoke to Dr. Miller, who gave this in a sermon in Vallejo, CA, circa 2014.

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
