# Peer review of "A Comedian in the Pulpit: Empowering the Use of Humor in Preaching"

_religions, doi:10.3390/rel14091155_

Round 1

Reviewer 1 Report

A Comedian in the Pulpit is interesting topic that could be improved.

For example, in section 5, analyzing or describing humor, should or could contain samples of sermons embedded in humor. Essay could be more ethnographical were the author to give at least one sample of satire or humor in preaching.

Suggestions to improve the language and presentation of the essay

Rephrase or turn phrases into sentences ll. 77-80;

Take another look at the phrases/sentences, ll. 81-89; ll. 137-138; 225; 308-310.

Make corrections: line 75 – “This is not just” not “This I not just”;

l. 293, “than” not “that”; l. 327, “ruse” not “rouse”;

Author Response

I looked over the issues that were presented and made the necessary adjustments. Thank you for your feedback. 

Reviewer 2 Report

This is a very important subject and essay! The paper is quite convincing as a programmatic and historical informed essay rather than a empirical research. Hence the empirical methods are not central and could be even reduces without loosing substance. Very conclusive is the systematic reflection of humor and the historical reconstruction of the relationship of humor / trickster / preaching als slavery, comparing with the function of humor for jews.
In the introduction humor in preaching seems to be the central key for theological and homiletical quality, neglecting the ambivalent and even risky aspects of humor and mockery for preaching the word of God.

I recommend the publication the paper as a programmatic and historicaly informed essay and not as an empirical research.

Perhaps the author could integrate the following book:

Charles L. Campbell / Johan H. Cilliers: Preaching Fools. The Gospel As a Rhetoric of Folly

Author Response

Thank you for your feedback. I recently acquired the book Preaching Fools, but this was after I completed the paper. My intention is to include this book in future work. 

Reviewer 3 Report

Submitted to the journal Religions, the article presents a little-known, especially in Europe, connection between humor and the culture of African-Americans and also the possibility of using humor in preaching in the African Methodist Episcopal Zion Church. The issues addressed in the article are at the intersection of ethnography and theology of preaching. I make only three comments.

1. The title of the article is too general. He - in my opinion - demands a subtitle presenting the essence of the study conducted.

2. Abstract is also too general, short and imprecise. I suggest supplementing it with the essential elements of the scientific study undertaken. Admittedly, the scientific problem was expressed in the question: "Does humor have a place in that sacred moment?" However, the author has not presented the main elements of the research procedure, especially the assumed objectives of the study, the scientific sources used, and the stages of the research procedure. It would be good to specify more precisely the field(s) of study and the modus of the methodology used.

3. The author, in section 5.2 Assessing - Analyzing, cites the results of a survey of AMEZ preachers, but does not disclose how many preachers filled out the survey, when and in which country/countries this was done. Scientific soundness demands that this data be supplemented. Moreover, since the empirical research was conducted among AMEZ preachers, it makes no sense to extend the conclusions to the broad black church. In the presentation of the historical and cultural context of the issue under study, one can write about the black church, but the conclusions should focus on the AMEZ.

Author Response

Thank you for your Feedback. I updated the title, abstract, and conclusion to reflect the nature of the study better.